# Adolescent Intermittent Ethanol (AIE) Enhances the Dopaminergic Response to Ethanol within the Mesolimbic Pathway during Adulthood: Alterations in Cholinergic/Dopaminergic Genes Expression in the Nucleus Accumbens Shell

**DOI:** 10.3390/ijms222111733

**Published:** 2021-10-29

**Authors:** Sheketha R. Hauser, Patrick J. Mulholland, William A. Truitt, R. Aaron Waeiss, Eric A. Engleman, Richard L. Bell, Zachary A. Rodd

**Affiliations:** 1Department of Psychiatry, Institute of Psychiatric Research, Indiana University School of Medicine, Indianapolis, IN 46202, USA; shhauser@iupui.edu (S.R.H.); btruitt@iupui.edu (W.A.T.); rwaeiss@iupui.edu (R.A.W.); eenglema@iupui.edu (E.A.E.); ribell@iupui.edu (R.L.B.); 2Department of Neuroscience, Charleston Alcohol Research Center, Medical University of South Carolina, Charleston, SC 29425, USA; mulholl@musc.edu

**Keywords:** adolescent alcohol, dopamine self-administration

## Abstract

A consistent preclinical finding is that exposure to alcohol during adolescence produces a persistent hyperdopaminergic state during adulthood. The current experiments determine that effects of Adolescent Intermittent Ethanol (AIE) on the adult neurochemical response to EtOH administered directly into the mesolimbic dopamine system, alterations in dendritic spine and gene expression within the nucleus accumbens shell (AcbSh), and if treatment with the HDACII inhibitor TSA could normalize the consequences of AIE. Rats were exposed to the AIE (4 g/kg ig; 3 days a week) or water (CON) during adolescence, and all testing occurred during adulthood. CON and AIE rats were microinjected with EtOH directly into the posterior VTA and dopamine and glutamate levels were recorded in the AcbSh. Separate groups of AIE and CON rats were sacrificed during adulthood and Taqman arrays and dendritic spine morphology assessments were performed. The data indicated that exposure to AIE resulted in a significant leftward and upward shift in the dose-response curve for an increase in dopamine in the AcbSh following EtOH microinjection into the posterior VTA. Taqman array indicated that AIE exposure affected the expression of target genes (*Chrna7*, *Impact*, *Chrna5*). The data indicated no alterations in dendritic spine morphology in the AcbSh or any alteration in AIE effects by TSA administration. Binge-like EtOH exposure during adolescence enhances the response to acute ethanol challenge in adulthood, demonstrating that AIE produces a hyperdopaminergic mesolimbic system in both male and female Wistar rats. The neuroadaptations induced by AIE in the AcbSh could be part of the biological basis of the observed negative consequences of adolescent binge-like alcohol exposure on adult drug self-administration behaviors.

## 1. Introduction

Despite multiple intervention programs, the overall drinking rate of adolescence in the US has remained relatively constant for decades [1,2]. Adolescent/young adult binge drinking has become more prevalent in younger adolescents and there has been an increase in the overall rate of binge drinking during the transition from late adolescent/young adulthood into adulthood [3]. The increase in the occurrence of binge drinking in adolescents/young adults has required the further classification of high-intensity, and extreme-intensity adolescents that have recently emerged [4,5]. Almost a third of young adults (21–26) in the US report recent bouts of binge-drinking (30%), 11% report high-intensity drinking, and 5% report extreme-intensity drinking [2]. Binge-drinking in adolescence is also associated with the likelihood of polydrug use/abuse during adolescence [6].

There are many things that predispose an individual to develop AUD during a lifetime. There is strong evidence for a genetic predisposition to develop AUD [5]. Individual behaviors can also amplify the risk for developing AUD. There is a positive correlation between age of first drink and harmful consequences of alcohol use during adulthood (increase alcohol involvement, rate of heavier drinking bouts, arrests for driving with ability impaired, and rates of AUD during adulthood; [7,8]). Individuals who initiate alcohol use prior to the age of 15 have a 1.3 to 1.6 times increased rate of AUD during adulthood compared to individuals who initiate alcohol consumption after the age of 15 [9].

Animal models that parallel adolescent human drinking have been developed. A concern of a significant portion of preclinical models is that adolescent alcohol exposure does not surpass pharmacologically relevant blood alcohol levels (>30 mg%). Other models employed are stressors (vapor chambers) or could have long term physical consequences (repeated injections of large volumes of fluid in adolescent rats). The Alcohol Intermittent Exposure (AIE) model was created with the goal of providing consistent binge-like alcohol exposure (>80 mg%) in adolescent rats [10]. However, the actual exposure obtained by what should be the standard AIE protocol (gavage and 4 g/kg) produces BEC levels closer to high-intensity drinking [10]. 

Adolescence is a critical time of neurodevelopment. The adolescent brain undergoes periods of rapid growth, structural reorganization, and neurogenesis/neuronal pruning [11]. Alcohol (and other drugs of abuse) alters the normal neuronal developmental processes. Replicated findings indicate that alcohol exposure/consumption during adolescence results in a hyperdopaminergic response to stimuli, reduction in choline acetyltransferase (ChAT), alterations in the neuroimmune system, and non-specific modulation of epigenetic factors [12,13,14,15]. 

The mesolimbic dopamine system mediates several motivated behaviors (sex, consummatory behaviors, maternal behaviors, and other response-contingent behaviors; [16,17]). A hyper-responsive dopaminergic system would predispose an organism to have an altered threshold for stimuli to establish motivated behaviors (e.g., reinforcement). Therefore, organisms with a hyper-dopaminergic system would respond to subthreshold stimuli and develop behaviors to obtain these subthreshold stimuli that would not be observed in organisms with ‘normal/unmodified’ dopaminergic systems. 

Consistent preclinical findings have reported that adolescent EtOH consumption [18] or peripheral administration of EtOH [19], increases basal dopamine levels of dopamine reuptake in the AcbSh during adulthood. Comparable exposure to EtOH during adulthood does not produce similar effects [20,21]. Selectively bred alcohol-preferring (P) rats consuming alcohol during adolescence (BECs approximately 60 mg%) results in a leftward and upward shift in the dose-response curve for EtOH self-administration directly into the posterior VTA [pVTA; 22]. Using the AIE procedure, Wistar and P rats exposed to binge-like levels of alcohol during adolescence produced a comparable leftward and upward shift to the dose-response curve for ethanol self-administration directly into the pVTA. Divergence in response to stimuli is indicated by the observation that the concentration of ethanol microinjected into the pVTA required to evoke DA release in the AcbSh during adulthood is reduced in P rats that voluntarily consumed alcohol during adolescence [22].

The behavioral effects of a hyper-dopaminergic system produced by adolescent binge-like alcohol exposure is suggested by multiple reports. Voluntary adolescent EtOH consumption during adolescence results in more ‘risky’ choices (and a comparable greater DA release in the AcbSh following ‘risky’ decisions) during adulthood [23]. Voluntary alcohol consumption in P rats enhances the acquisition of EtOH self-administration, delays extinction of established EtOH self-administration, enhances alcohol relapse drinking, and increases context- and cue-induced EtOH-seeking during adulthood [24]. These altered behaviors were not observed following comparable voluntary alcohol consumption in adult P rats [25]. Voluntary alcohol consumption in adolescent P rats increases the reward valence (higher breakpoint values) during adulthood [26]. Further research indicated that these findings were specific to alcohol because comparable voluntary alcohol consumption in adolescent P rats did not alter adult saccharin self-administration [26]. 

The effects of adolescent alcohol exposure are time-dependent (early versus late adolescent exposure) and sex-dependent [27]. AIE exposure results in an increased in adult levels of anxiety-like behaviors, which can be reversed by oxytocin and vasopressin agents [28]. In mice, AIE exposure increased acquisition of EtOH drinking and relapse drinking in males but not females [29]. AIE produces sex-dependent alterations in glutamate transmission (females only) and kappa opioid receptor function (males only) in the basolateral amygdala in Sprague-Dawley rats [30]. Therefore, there is a need to examine any effects of adolescent exposure in both sexes.

The enhanced drug self-administration observed in adults following adolescent EtOH exposure is thought to be mediated, in part, by a series of biological events: (1) alcohol acts on receptors or ion channels (perhaps more targets) to alter activity within the adolescent brain, (2) activation of these EtOH targets induce production of epigenetic factors that persistently alter the genetic expression of specific proteins, alterations in protein structures and other processes, (3) the compromised adult brain (containing the neuroadaptations produced by adolescent alcohol exposure) reacts differently to alcohol and other drugs of abuse which increases the propensity or these individuals to consume drugs [31]. Alterations in the mesolimbic dopamine system (pVTA projections into the AcbSh) could be the biological basis for the increase in the propensity to consume alcohol and other drugs of abuse during adulthood [12,22,32,33]. Therefore, it is critical to examine the effect of adolescent EtOH exposure on adult response to EtOH within the mesolimbic dopamine system during adulthood. 

The current experiments isolated the effects of EtOH within an in vivo mesolimbic dopamine system by microinjecting EtOH into the posterior VTA and recording dopamine (and at times glutamate) in the AcbSh. The hyperdopaminergic consequence of adolescent alcohol exposure could also involve alterations in post-synaptic responses in the AcbSh in response to exposure to alcohol during adulthood. Therefore, alterations in gene expression of dopaminergic and cholinergic factors and dendritic spine morphology in the AcbSh was performed in adolescent exposed and naïve Wistar male and female rats. Although alcohol (EtOH) has not been shown to act directly on epigenetic markers, there has been some reports indicating that blocking epigenetic factors may have therapeutic efficacy to ‘normalize’ adult brains that were exposed to EtOH during adolescence [15]. The current study examines the ability of the HDACII inhibitor Trichostatin A (TSA) to prevent adolescent alcohol induced enhancement in the reinforcing properties of EtOH within the posterior VTA and enhanced dopamine release in the AcbSh following EtOH microinjection into the posterior VTA.

## 2. Results

### 2.1. Experiment 1: Effects of AIE Treatment on the Neurochemical Response to EtOH within the Mesolimbic Dopamine System

The overall analysis indicated a significant ‘Adolescent Exposure’ × ‘EtOH Concentration’ × ‘Sample Time’ interaction (F_32,418_ = 2.54; *p* < 0.0001). Significant interaction terms are decomposed (hold one value constant while perform similar analysis with that factor removed) to help define the basis for the interaction term. The significant 3-way interaction term was decomposed by holding ‘Adolescent Exposure’ constant (performing separate analysis in CON and AIE groups). In the CON group (Figure 1; left panel), there was a significant ‘EtOH Concentration’ × ‘Sample Time’ interaction term (F_32,202_ = 2.25; *p* < 0.001). The significant 2-way interaction term was decomposed by holding ‘Sample Time’ constant (performing individual ANOVAs at each time point). There were significant ‘EtOH Concentration’ group differences during the 1st and 2nd time point following EtOH microinjection into the posterior VTA (F_4,41_ values > 12.5; *p* values < 0.001). Post-hoc comparisons (Tukey’s b) indicated that CON rats microinjected with 150 and 200 mg% had significantly higher DA levels in the AcbSh than all other groups.

AIE rats were more responsive for the ability of EtOH microinjected into the posterior VTA to stimulate DA release in the AcbSh. In AIE rats, there was a significant ‘EtOH Concentration’ × ‘Sample Time’ interaction term (F_32,216_ = 3.84; *p* < 0.0001). Individual ANOVAs indicated significant effects of ‘EtOH Concentration’ during the 1st–4th time points following EtOH microinjection into the posterior VTA (F_4,45_ values > 13.48; *p* values < 0.001: Figure 1; right panel). During the 1st injection period, AIE rats microinjected with 75, 150 or 200 mg% EtOH (each of these EtOH concentration was significantly different from each other) into the posterior VTA had significantly higher DA levels in the AcbSh than aCSF and 50 mg% injection groups. Similar results were observed during the 2nd (150 and 200 mg% > 75 mg% > 50 mg% and aCSF) and 3rd (75, 150, and 200 mg% > aCSF and 50 mg%) 20-min time period following EtOH microinjection into the posterior VTA. During the 4th time point after EtOH microinjection into the posterior VTA the two highest concentrations of EtOH (150 and 200 mg%) had significantly higher levels of DA in the AcbSh compared to all other groups.

The significant 3-way interaction term can also be decomposed by holding ‘EtOH Concentration’ constant (comparing CON/AIE rats at each EtOH concentration tested). There was significant ‘Adolescent Exposure’ × ‘Time Sample’ interactions (F_8,148–168_ values > 2.89; *p* values < 0.001) in rats microinjected with 75, 150, and 200 mg% into the posterior VTA. These two-way interactions were similarly decomposed by examining the effect of ‘Adolescent Exposure’ at each sample time point in the 75, 150, and 200 mg% groups. It is critical to state that subsequent ANOVAs were performed with an Independent Variable (Adolescent Exposure) with only two levels. Therefore, although we are reporting F values, all F tests performed with only two levels of comparisons are t-tests in statistical reality (see Keppel, 1973 for an outstanding derivative-based analysis of this statement). DA levels in the AcbSh was elevated during the 1st–3rd 20-min time periods in AIE rats administered 75 mg% EtOH into the posterior VTA compared to comparable CON rats (F_1,16_ values > 19.4; *p* < 0.001). In rats microinjected with 150 or 200 mg% EtOH into the posterior VTA, DA levels in the AcbSh were greater in AIE rats compared to CON rats during the 1st–4th period (F values > 7.62; *p* values < 0.01). 

### 2.2. Taqman Array Assessment of DA and Acetylcholine Receptors in the AcbSh Following AIE Treatment

Genetic expression was normalized to the expression of housekeeping genes *18S*, *Gapdh*, *Ubs*, and *Ywhaz*. Of the 42 DA and AchR related genes for analysis selected, 10 were flagged by initial Taqman procedure indicating insufficient genetic expression for reliable detection. These genes included *Cdnf*, *Chrna1*, *Chrna9*, *Chrna10*, *Chrnd*, *Chrne*, *Chrng*, *Dbh*, *Drd4*, and *Hrt3b*. Gene expression was contrasted to expression levels recorded in male CON rats. The log^10^ values for all non-significant validated genes are included in Table 1 (fold change compared to male CON rats). Conversion of the data to the log^10^ values reduced variability and allowed for accurate assessment of the magnitude of the effect in all groups. Data conversion increased the power estimate for all significant gene analyses. Cohen h tests indicate that for gene expression changes that approached significance (0.06–0.20) there was moderate power and the observed effects sizes were small (0.1). In contrast, for the *Chrna7* analysis the power estimate was high (0.83) and the effect size was large (0.87). For the expression for 4 genes (Figure 2) there was a significant effect of Adolescent Exposure (F_3,15_ values > 6.57; *p* values < 0.002). Specifically, AIE treatment results in a persistent decrease in the expression of *Chrna5* and persistent increase in the expression of *Chrna7*, *Chrnb4*, and *Impact* (*p* values < 0.006). There were significant Adolescent Exposure x Sex interactions for the expression of 7 genes (F_3,15_ values > 8.03; *p* values < 0.001; Figure 3). The significant interaction terms were often produced by AIE influencing the gene expression in only male rats. Specifically, the expression of 3 genes (*Drd1*, *Drd3*, and *Htr3a*) was reduced in AIE males, but no effect in female AIE rats. In contrast, the expression of *Chrm3* and *Slc22a3* was increased in AIE males, but no effect in female AIE rats. There was an innate sex difference for the gene expression of *Chrna2* (85% suppression in females compared to males), and AIE exposure increased the rate of the expression of *Chrna2* in both sexes (greater increase in male rats). AIE exposure increased the expression of *Chrm2* in male rats, but there was an innate higher level of expression of this gene in CON female rats.

### 2.3. Effects of AIE Treatment on Dendritic Spine Morphology in the AcbSh

In general, there were no significant effects of AIE exposure on dendritic spine density or morphology in the AcbSh (Figure 4). Initial analyses were conducted on the following dependent measures; spine length, spine mean diameter, spine volume, dendritic spine density, spine neck volume, spine neck mean diameter, and spine neck length. All analyses failed to observe significant results (*p* values > 0.22). A multifactor ANOVA was performed to determine if there was a general group differences across multiple dependent measure. This analysis also failed to observe significant effects (*p* values > 0.13).

### 2.4. Effects of Administration of Trichostatin on AIE-Induced Enhancement of EtOH Microinjected into the Posterior VTA on Dopamine and Glutamate Levels in the AcbSh

The overall 4-way analysis indicated that there was no effect of TSA pretreatment/treatment on alterations in DA levels in the AcbSh induced by microinjections of EtOH into the posterior VTA (*p* values > 0.87). There was a significant ‘Adolescent Exposure’ × ‘EtOH Concentration’ × ‘Sample Time’ interaction (F_24,358_ = 2.61; *p* < 0.0001). Identical to the results of Experiment 1, AIE exposure produced a persistent alteration in the posterior VTA that resulted in a leftward and upward change in the EtOH dose-response curve (Figure 5). Succinctly, in CON rats, microinjection of 150 mg%, but not 75 mg%, EtOH into the posterior VTA increased DA levels in the AcbSh, and TSA treatment had no effect on any parameter (F_24,170_ = 1.97; *p* < 0.001; Tukey’s b post-hoc comparisons). In AIE rats, administration of 75 and 150 mg% EtOH into the posterior VTA stimulated DA release in the AcbSh, and TSA treatment had no effect on any parameter (F_24,188_ = 3.77; *p* < 0.0001; Tukey’s b post-hoc comparisons). 

Recent published findings have indicated that co-administration of EtOH and nicotine into the posterior VTA (but not comparable administration of EtOH or nicotine) results in simultaneous release of DA and glutamate in the AcbSh. The overall analysis indicated that microinjection of only EtOH into the posterior VTA failed to alter glutamate levels in the AcbSh independent of AIE exposure and/or TSA pretreatment/treatment (Figure 6; all *p* values > 0.72).

### 2.5. Effects of Administration of Trichostatin a on AIE-Induced Enhancement of EtOH Reward in the Posterior VTA

Overall, the data indicate that pretreatment/treatment of the posterior VTA with TSA had no effect on EtOH self-administration or AIE-induced enhancement of EtOH reward in the posterior in Wistar or tP rats. In Wistar rats (Figure 7), the overall statistical analysis indicated only a significant three-way interaction ‘Adolescent Exposure’ × ‘EtOH Concentration’ × ‘Session’ (F_6,284_ = 4.19; *p* < 0.0001). There were no significant effects of ‘Sex’ or ‘HDAC Exposure’ on any statistical assessment (*p* values > 0.41). The lack of a ‘Sex’ effect resulted in data from both males and females being collapsed. In all Wistar rats, the significant three-way interaction term was decomposed by holding ‘Adolescent Exposure’ and ‘EtOH Concentration’ constant. In CON Wistar rats, there was a significant ‘EtOH Concentration’ × ‘Session’ interaction (F_6,140_ = 4.74; *p* < 0.0001). In CON Wistar rats allowed to self-administer 75 mg% EtOH directly into the posterior VTA there was no effect of ‘Session’ (*p* = 0.61). This result indicated that active lever responding in these rats did not differ between EtOH and aCSF self-administration (indicating a lack of EtOH reinforcement). In CON Wistar rats allowed to self-administer 150 mg% directly into the posterior VTA, there was an effect of ‘Session’ (*p* = 0.006). Post-hoc comparisons (two-sided *t*-tests) indicated that active lever responding during sessions 1–4 and 7 was significantly greater than that observed during aCSF substitution (sessions 5 and 6; *p* values < 0.02). In AIE Wistar rats there was also a significant ‘EtOH concentration’ × ‘Session’ interaction (F_6,144_ = 2.87; *p* < 0.001). In AIE Wistar rats, for both the 75 and 150 mg% EtOH concentrations there was a significant effect of Session (*p* values < 0.003) and post-hoc comparisons indicated that active lever responding was greater when EtOH was available in the infusate (sessions 1–4 and 7; *p* values < 0.001) then during aCSF extinction training (sessions 5 and 6). Contrasting active lever responding between CON and AIE rats indicated that AIE rats self-administering 75 mg% EtOH directly into the posterior VTA responded more than similar CON rats during all sessions EtOH was available in the infusate (sessions 1–4 and 7; *p* values < 0.007). In Wistar rats self-administering 150 mg% EtOH, AIE rats responded more on the active lever during sessions 2–4, and 7 compared to CON rats (*p* values < 0.02). Again it is worth noting that ‘HDAC Exposure’ and ‘Sex’ had no effect on EtOH self-administration behaviors therefore means were collapsed for the analyses.

In tP rats, the overall analysis indicated only a significant effect of ‘Session’ (Figure 8; F_6, 52_ = 3.82; *p* < 0.01). There was no effect of ‘Sex’ or ‘HDAC Exposure’. Therefore, the data replicated the Wistar data in indicating that pretreatment/treatment with TSA directly in the posterior VTA had no effect on the rewarding property of EtOH or AIE-induced enhanced EtOH reward in that brain region.

## 3. Discussion

The results of the current experiments indicate that binge-like exposure to EtOH during adolescence in male and female Wistar rats results in persistent neuroadaptations in the mesolimbic dopamine system (pVTA). The hyperdopaminergic state during adulthood in AIE rats is indicated by the greater sensitivity (lower EtOH concentrations needed to elicit a response) and enhanced responsivity (enhanced response to equivalent stimulus) in the ability of EtOH microinjected into the posterior VTA to stimulate DA release in the AcbSh (Figure 1). The enhancement in activity within the mesolimbic DA system produced by AIE exposure was combined with specific alterations in the genetic expression of dopaminergic and cholinergic factors within the AcbSh (Figure 2 and Figure 3). Therefore, AIE exposure is associated with persistent alterations in the input into the AcbSh (posterior VTA DA) and possibly the interpretation of the altered input (different receptor expression levels). It is likely that the increase in adult alcohol consumption observed in adolescent alcohol-exposed rats is dependent upon alterations in dopaminergic inputs from the VTA into the AcbSh and the translation of that input into the AcbSh into feed-forward outputs to other critical brain structures [12,31].

Previous research indicated that voluntary EtOH consumption or experimenter administered EtOH during adolescence resulted in a persistent hyperdopaminergic system during adulthood. In P rats voluntary EtOH consumption during adolescence increased the sensitivity to the reinforcing effects of EtOH in the pVTA and an increased responsivity (leftward and upward shift in the dose-response curve) in the ability of EtOH microinjected into the pVTA to stimulate DA release in the AcbSh [22]. The current data extend the published findings by indicating that experimenter administered binge-like EtOH exposure during adolescence in Wistar rats produces comparable results (Figure 1, Figure 5 and Figure 6) and thus provides further validation of the AIE model. The consistent finding that adolescent EtOH exposure results in a hyper-dopaminergic state during adulthood is indicated by the findings that systemic administration of EtOH during adolescence results in higher basal DA levels in the Acb [19,20,21] or DA turnover (increase DA neurotransmission) in the dorsal striatum [34]. An increase in DA neurotransmission was also observed in adult P rats following adolescent EtOH consumption (increased DA clearance) within the Acb [18]. Although the enthusiasm for DA focused research is currently passé, it is important to state the consistent, influential effects that adolescent alcohol exposure has on the mesolimbic DA and the importance that the system has on regulating multiple EtOH-related behaviors.

Similar to the persistent hyperdopaminergic state during adulthood, adolescent exposure to binge-like EtOH has been repeatedly shown to reduce ChAT in multiple brain regions during adulthood [10,15,35]. Reduction in ChAT levels in adulthood is also observed following adolescent exposure to other drugs of abuse [36]. In general, the effects of the reduction in ChAT have been not properly followed-up. Conceptually, the reduction in ChAT levels produced by adolescent binge-like EtOH exposure should be associated with an increase in acetylcholine receptors. Taqman array analyses indicate that adolescent alcohol exposure results in a decrease in α5 and an increase in α7 nicotinic acetylcholine receptors (NAchRs) in the AcbSh (Figure 2) and the posterior VTA [12]. Voluntary EtOH consumption in adolescent P rats reduced the concentration of nicotine microinjected into the pVTA required to stimulate DA release in the AcbSh [37,38]. Furthermore, adolescent voluntary binge-like EtOH drinking in P rats is associated with an increase in α7 labeling and protein levels in the pVTA [37]. 

The importance of the α5/α7 NAchRs in mediating the effects of adolescent EtOH exposure on subsequent adult EtOH/drug intake needs to be fully developed. The α3, α5, and β4 NAchRs expression have been associated with differential risk for the development of nicotine dependence (ND) and alcohol use disorder (AUD) [39]. Activation of the α7 receptor has both fast and sustained actions within the VTA and AcbSh [40]. Specifically, the α7 receptor is both a fast action/quickly desensitized ligand-gated ion channel (increase Ca^2+^ influx) and a long-acting G-protein coupled receptor (Gαq which promotes the sustained release of Ca^2+^ in the neuron; [41]). Activation of the α7 receptor in the VTA also stimulates DA neurons [42]. Site-specific administration of α7 agonists induces immediate early gene responses in the AcbSh and not the Acb Core or other surrounding brain structures [43]. The increase in the expression of *Chrna7* in the AcbSh (Figure 2) is most likely expressed in presynaptic inputs emerging from cortical regions and reflect alterations in specific neuronal pathways [44]. These α7 NAchR pathways within the AcbSh have also been shown to mediate cannabinoid self-administration [45]. Thus, upregulation of *Chrna7* expression within the mesolimbic DA pathway may be part of the biological basis for the cross-sensitivity effects of adolescent EtOH exposure on the effects of other drugs of abuse during adulthood [37,38,46,47]. However, alteration in gene expression for the multiple DA and NAchRs in the AcbSh is not associated with parallel adaptations in dendritic spine density (Figure 4).

The analysis also indicated significant changes in the gene expression of targets of interest. There were innate sex differences in the expression of *Chrna2* and *Chrm2* (Figure 3). *Chrna2* has been postulated as a candidate gene influencing antisocial drug dependency in adolescence [48]. AIE effects on the expression of some genes were specific to males (*Drd1*, *Drd3*, *Htr3a*, *Slc22a3*, *Chrm3*). Therefore, the current data set parallels other publications indicating that males may be more responsive to adolescent alcohol exposure [37]. The gene that codes the Impact receptor (ancient and conserved) was increased in both sexes. The Impact receptor is a unique receptor that has similar properties to cys-loop receptors (α7 and 5HT3). The Impact receptor appears to be highly responsive to exposure to alcohol because past reports have also indicated significant alterations in the genetic expression of this receptor [49,50]. 

TSA is both a toxin and a mutagenic agent. Similar to all HDAC inhibitors, TSA results in embryonic death and reduction in life expectancy because of the accumulation of toxins within organisms [51,52]. Exposure to HDAC inhibitors (TSA, vorinostat, and others) cause single nucleotide variants (SNVs) in multiple cell types and results in damage to DNA that is comparable to cancer-inducing stimuli [53]. Administration of TSA at levels used in alcohol studies ([15,54,55], Figure 5, Figure 6 and Figure 7) results in elevated structural chromosomal aberrations (increase or decrease number of chromosomes) in a number of cell types [56]. In addition, the administration of TSA is associated with increases in numerous measures of DNA damage (non-banding metaphase chromosomes, kinetochore-antibody micronucleus proteins, levels of aneuploidy, and other factors [56]). Although TSA is approved for the treatment of some types of cancer, the possibility of TSA-induced cancers and other toxin-related, mutagenic-related side-effects should greatly limit the use of HDAC inhibitors to treat only dire human conditions [53].

Epigenetic factors are critical to the development of persistent neuroadaptations produced by adolescent alcohol exposure that increases the likelihood of AUD and drug use during adulthood [57]. However, it is unlikely that epigenetic factors mediate the acute effects of EtOH. The neurological sites that EtOH acts directly upon include the cys-loop ligand-gated ion channel (NAchRs, 5HT3, GABAA, glycine, Impact), VTA DA neurons, NMDA receptor, and specific ion channels [58,59,60,61]. AIE exposure, or voluntary adolescent alcohol consumption, increases neurotransmitter receptors within the mesolimbic dopamine system ([12,37], Figure 2 and Figure 3). Brief interventions with toxic HDAC inhibitors are not going to eliminate the alterations in neurotransmitter receptors levels within the mesolimbic dopamine system. Furthermore, critical studies determining if HDAC inhibitors reverse AIE-induced alterations in neurotransmitter receptor levels in adulthood has not been conducted. The current data indicates that chronic exposure to TSA failed to alter the hyperdopaminergic systems induced by voluntary adolescent alcohol consumption or AIE exposure (Figure 5, Figure 6, Figure 7 and Figure 8). The mesolimbic DA system is a critical neurocircuitry mediating EtOH reward, motivated behaviors to obtain EtOH (EtOH-seeking), and EtOH self-administration. The failure of TSA to mediate AIE-induced enhancement of EtOH actions within the mesolimbic DA system during adulthood greatly reduces the enthusiasm for using these aversive compounds to address the enduring effects of adolescent binge-alcohol consumption. 

In contrast, the α7 NAchR appears to be a highly viable target to prevent/reverse the effects of adolescent binge-alcohol exposure. Administration of the α7 NAchR negative allosteric modulator (NAM) dehydronorketamine 2 h before adolescent binge-EtOH exposure during adolescence, prevented the increase in the acquisition of EtOH consumption and the enhancement of relapse EtOH consumption during adulthood in AIE exposed rats [31]. Daily administrations of an α7 NAchR agonist (AR-R1779) during early adolescence (only 6 days of exposure) resulted in adult EtOH consumption similar to rats given AIE exposure (enhanced acquisition and relapse [31]). Our results were conceptually replicated with the recent report that pretreatment with the cholinesterase inhibitor galantamine (countering the activity of the α7 receptor) prevented ABAE-induced increase in the expression of TLR4, RAGE, HMGB1, and pNF-κB p65 during adulthood [62]. Donepezil can reverse AIE-induced decreases in dendritic spine density and expression of the *Fmr1* gene in the hippocampus [14]. The effects of Donepezil (e.g., neuroprotection against toxins, regulation of the neuroimmune system, alterations in the genetic expression of other systems) are primarily thought to be through increased activity of the α7 system [63,64]. Given the multiple lines of evidence of neuroadaptations in the α7 NAchR following adolescent binge EtOH exposure (Figure 2, [12,37]) and the potential therapeutic effects of α7 NAchR agents to prevent or reverse the effects of adolescent binge EtOH exposure, the α7 NAchR must be consider a prime target for future studies attempting to address the consequences of adolescent binge EtOH exposure.

Recently published data have indicated that co-administration of EtOH + nicotine into the posterior VTA induces the releases of both DA and glutamate in the AcbSh [38]. Similarly, only EtOH + nicotine co-administration into the posterior VTA results in increases in BDNF expression in the AcbSh and subsequent enhancement in the reinforcing properties of EtOH in the AcbSh [38]. The original hypothesis of Experiment 4 was that AIE exposure would result in neuroadaptations within the posterior VTA that would fundamentally change the downstream neurochemical (AcbSh) response (increases in both DA and glutamate) to EtOH administered directly into the posterior VTA. The data do not support that hypothesis (Figure 6). Thus, the data reinforces the repeated findings that concurrent exposure to EtOH and other drugs results in unique responses and neuroadaptations within the brain [38,65,66,67]. 

Overall, the current results suggests that adolescent binge-like EtOH exposure produces persistent alterations within the mesolimbic DA pathway (hyperresponsive dopaminergic system). Given the overwhelmingly consistent findings, it is imperative that cholinergic interventions to counter/block the adult consequences of adolescent binge-like EtOH exposure be further developed. Future research will need to examine if pretreatment with α7 NAMs or galantamine could prevent the significant findings reported in this manuscript. Ethically, it would be better to prevent the deleterious consequences of a normal adolescent behavior (alcohol consumption) than to attempt to treat an individual that has been negatively impacted by a process that could have been prevented. Currently, it is perhaps the most intensive period of research examining adolescent alcohol drinking, it is critical that possible interventions be developed and becoming reductionist traveling down spurious rabbit holes be avoided.

## 4. Methods

### 4.1. Subjects

A single generation colony of Wistar rats has been established at Indiana University School of Medicine (IUSM; Indianapolis, IN, USA). The Taconic P (tP) rat colony was maintained at Indiana University (breeding and rearing in same building as experiments conducted). Recently, this valuable rat line (tP) was lost because of funding issues. Breeding is done within the facilities in order to maintain chain of control of subjects and to prevent stress from shipping adolescents. 

### 4.2. Adolescent Intermittent EtOH (AIE) Protocol

The rats were treated with the proposed standard NADIA AIE protocol. On post-natal day (PND) 21 rats were randomly assigned to either binge (AIE) EtOH exposure or water (Control Intermittent Exposure, CON) only. AIE rats were treated with 4 g/kg EtOH (25% *v*/*v*, i.g.) four times a week during adolescence (PND 28–48). The CON group received comparable treatment with water. EtOH and water was administered through oral gavage. All rats were group housed during adolescent treatment and until the onset of experiments PND 90.

### 4.3. Microinjection-Microdialysis Protocol

The microinjection-microdialysis procedure was carried out as described previously [35,65,68]. Following PND 90, rats were stereotaxically implanted with two ipsilateral guide cannulas in the right hemisphere. While under isoflurane anesthesia, a 22-guage microinjection cannula (Plastics One, Inc., Roanoke, VA, USA) was aimed 1.0 mm above the posterior ventral tegmental area (pVTA). An 18-guage microdialysis cannula was also implanted 3.0 mm above the NAc shell. The coordinates for the pVTA were AP −5.6 mm, ML +2.1 mm, DV −8.0 mm and the NAc shell coordinates were +1.5 mm, ML +2.0 mm, DV −5.3 mm). Both cannulas were implanted at a 10° angle and protected with stylets while no experiments were being carried out. Following surgery, animals were single housed in shoebox cages and allowed at least one week of recovery. During this time, the animals were habituated to the experimental housing and handled daily.

Loop-style microdialysis probes were constructed as previously described [68,69]. Probes were manufactured with an active length of 2.0 mm from regenerated cellulose Spectra/Por (Spectrum Laboratories, Inc., Rancho Dominguez, CA, USA) hollow fiber tubing with an inner diameter of 200 µm and molecular weight cut-off of 13 kDa. One day prior to testing, animals were placed under isoflurane anesthesia and the microdialysis probes were inserted into the NAc shell by extending 3.0 mm below the guide cannula. The microinjection-microdialysis procedure was carried out the following day.

Experiments were performed in awake freely moving animals. Subjects were placed in the experimental chambers and the microdialysis probes were connected to a Harvard pump (Harvard Apparatus, Holliston, MA, USA) used to continuously perfuse the NAc shell with artificial cerebrospinal fluid (aCSF) at a rate of 1 µL/minute throughout the experiment. Microdialysis aCSF was composed of 140.0 mM NaCl, 3.0 mM KCl, 1.2 mM CaCl_2_, 2.0 mM Na_2_HPO_4_·7H_2_O, and 1.0 mM MgCl_2_ with a pH 7.2 to 7.4. Following a 90-min washout period, samples were collected in 20-min intervals beginning with five baseline samples.

Passive microinjections were carried out with an electrolytic microinfusion transducer (EMIT) system as described previously [12,70,71,72]. Briefly, subjects received 30 pulse injections over a 10-min period designed to emulate intracranial self-administration levels. Each pulse injection infused 100 nL of solution over 5 s and was followed by a 15-s timeout period for a total injection volume of 3 µL.

After the microinjection challenge, six additional 20-min samples of dialysate samples were collected into tubes containing five µL of 0.1 N perchloric acid. All samples were immediately frozen on dry ice and stored at −80 °C until analysis for dopamine content with high performance liquid chromatography (HPLC).

### 4.4. Intracranial Self-Administration (ICSA) Apparatus

The ICSA test chambers have been extensively described in the past [12,70,71,72]. Briefly, an electrolytic microinfusion transducer (EMIT) system controlled the delivery of assigned infusate into the subject via calibrated pulses of current. Depression of the active lever initiated a 5-s infusion current of 200 µA, resulting in rapid generation of hydrogen gas, increasing the pressure inside the airtight cylinder, pressing a 100 nL bolus of infusate out through the injection tip. Between infusions, current returned to the low quiescent state.

During each infusion, there was a 5-s time-out period where bar-press responses were recorded yet resulted in no further infusion. In this time-out, the house light and a red cue light were extinguished, while the green cue light over the active lever flashed in 0.5-s intervals. The assignment of active and inactive lever with respect to the left or right position was counterbalanced among subjects and remained the same throughout the experiment for each respective subject.

### 4.5. ICSA Procedure

Adolescent EtOH treatment occurred as described above. Food and water were available *ad lib* at all times, except during ICSA testing. ICSA was performed as previously described [12,70,71,72]. After postnatal day 75, the rats were implanted under isoflurane anesthesia with a guide cannula (22 gauge, Plastics One, Roanoke, VA, USA) stereotaxically aimed 1.0 mm above the pVTA. Coordinates were 5.8 to 6.1 mm posterior to bregma, 2.1 mm lateral, and 8.5 mm ventral from the surface of the skull at a 10 degree angle from the vertical (Paxinos and Watson, 1986). A place-holding stylet (28 gauge, Plastics One) extending 0.5 mm beyond the tip of the guide cannula was inserted at all times, except during test sessions. Subjects were single-housed post-surgery and allowed to recover for 7 days. Three days prior to testing, subjects were handled 5 min per day.

All infusates were prepared freshly on the day of the experiment. Artificial cerebrospinal fluid (aCSF) was used as the vehicle for ICSA infusions. This injection vehicle consisted of (in mM) 120.0 NaCl, 4.8 KCl, 1.2 KH_2_PO_4_, 1.2 MgSO_4_, 25.0 NaHCO_3_, 2.5 CaCl_2_, and 10.0 D-glucose. The solution was then sterilized with a syringe filter (pore size 0.2 µM) as previously described [12,70,71,72]. Ethyl alcohol (190 proof; McCormick Distilling Co., Weston, MO, USA) was dissolved in the vehicle solution to the correct concentration. When necessary, 0.5 N HCl was added to adjust the pH to 7.4 (±0.1).

ICSA was conducted similar to procedures previously described [12,70,71,72]. Briefly, subjects were brought to the testing room, the stylet was removed, and an injection cannula/infusate cylinder was affixed in place. The injection cannula extended 1.0 mm beyond the tip of the guide, into the pVTA. A single, noncontingent administration of infusate was given at the beginning of the session during this insertion procedure in order to prime the system. Test sessions occurred every other day. No operant shaping techniques were used. Active lever and inactive lever sides were counterbalanced between subjects, remaining the same for each individual rat. Within each 4 h session, responses on the active lever resulted in 5 s infusions on a fixed ratio 1 schedule of reinforcement. During infusion and time-out, responses on the active lever were recorded, but did not produce further infusions. Responses on the inactive lever were recorded but did not result in infusions at any time; these responses were used to index non-specific bar-pressing activity. During ICSA sessions 1 through 4 (acquisition), subjects received their respective dose of either the aCSF vehicle or EtOH. During ICSA sessions 5 and 6 (extinction), all subjects received aCSF vehicle only, and in session 7 (reinstatement), the original EtOH concentration was made available.

### 4.6. Effects of AIE Treatment on the Neurochemical Response to EtOH within the Mesolimbic Dopamine System

Overall, there were 88 (46 AIE and 42 CON) Wistar males used in this experiment. Rats were randomly assigned to have 0, 50, 75, 150, or 200 mg% EtOH microinjected into the posterior VTA. Group sizes were 8–10/group for AIE males, 8–9/group for AIE females, 7–9/group for CON males, and 7–9/group for CON females. Microinjection-Microdialysis experiments were conducted between PND 90–105.

### 4.7. Taqman Array Assessment of DA and Acetylcholine Receptors in the AcbSh Following AIE Treatment

A total of 23 Wistar rats were used in this experiment. Rats were exposed to the AIE (6 males/females) or CON (6 males/5 females) during PND 28–48. Rats were not exposed to any other manipulations prior to brains being harvested on PND 90. Samples were pooled from bilateral micropunches of the pVTA. The protocol for this procedure is detailed in published studies (Truitt et al., 2014). The TaqMan^®^ Low-Density Array (TMLDA) is a multiple RT-PCR assay that quantifiably examines mRNA levels. We have previously used the TMLDA assay to assess the effects of EtOH and NIC microinjected into the pVTA on gene expression in the AcbSh [73] and innate difference in gene expression within the pVTA in genetic models of alcoholism [65]. The current TMLDA (Applied Biosystems, Foster City, CA, USA) was a 48 gene array. The selected genes included 6 house-keeping genes and 37 genes associated with DA and Acetylcholine receptor systems (AchR: nAchR—nicotinic acetylcholine receptor). We also included other selected gene targets in the TLMDA (5 genes). The a7 NAchR is part of a cys-loop receptor family that includes the 5HT_3_ (*Hrt3b*a and *Hrt3b*) and Impact (Ancient and Conserved/Imprinted) receptor [74]. Originally, the 5HT_3_ receptor was classified as a NAchR because of a low affinity for acetylcholine [75]. Nicotine is not specific for NAchRs. In fact, nicotine binds at lower affinity to the 5HT_3_ receptor (Impact receptor has a similar binding profile) than any NAchR [75,76,77]. 

### 4.8. Effects of AIE Treatment on Dendritic Spine Morphology in the AcbSh

Neuronal labeling and morphological classification of dendritic spines on medium spiny neurons in the AcbSh were completed using previously reported methods (PMID: 25787124; 29336496). Adult Wistar rats (*n* = 43) were exposed to AIE (13 males/9 females) or CON (13 males/8 females) during PND 28–48 prior to brain harvesting on PND 90. Rats used for spine analysis were not exposed to other manipulations between PND 48 and 90. On PND 90, rats were anesthetized with urethane (1.2–1.5 g/kg, IP) and were perfused with 0.1 M phosphate buffer (PB) followed by 1.5% paraformaldehyde (PFA) in PB. Brains were blocked and post-fixed in 1.5% PFA for 1 h at room temperature. Brains were washed and shipped to the Medical University of South Carolina overnight in PB at 4 °C. Coronal sections (150 µm thick) were prepared using a vibratome. DiI-coated tungsten particles (1.3 µm diameter) were delivered to the slices using a modified Helio Gene Gun (Bio-Rad; Hercules, CA, USA) fitted with a polycarbonate filter (3.0 µm pore size; BD Biosciences; San Jose, CA, USA). Slices were left overnight at 4 °C in PB to allow the DiI to completely diffuse through labeled neurons and sections were post-fixed in 4% PFA for 1 h at room temperature. After mounting with Prolong Gold Antifade mounting media (Life Technologies; Carlsbad, CA), images (*n* = 137 dendritic sections) of distal sections of dendrites (>75 µm away from the soma) were acquired using a Zeiss LSM 880 confocal microscope (voxel size: 45 × 45 × 130 nm; 2 frame average) equipped with an Airyscan detector and a 63× oil immersion objective (Plan-Apochromat, Carl Zeiss, NA = 1.4, working distance = 190 µm). Raw Airyscan images were post-processed (Wiener filter-based deconvolution with pixel reassignment) in ZEN Black Software (version 14; Carl Zeiss) using the automatic filter strength setting. Imaris XT (version 9.1; Bitplane; Zurich, Switzerland) was used to generate a filament of the dendritic shaft and spines. Dendritic spines (*n* = 11,415 total spines) were identified using Imaris software and then classified into 4 categories (stubby, long, filopodia, and mushroom) based on the spine length and the width of the spine head and neck, where *L* is spine length, *D_H_* is spine head diameter, and *D_N_* is spine neck diameter. Long spines were identified as having a *L* ≥ 0.75 μm and < 3 μm, mushroom spines had a *L* < 3.5 μm, *D_H_* > 0.35 μm and a *D_H_* > *D_N_*, stubby spines had a *L* < 0.75 μm, and filopodia were identified as having a *L* ≥ 3 μm. Data on dendritic spine parameters were averaged for each dendritic section and were collated from the Imaris output via custom scripts written in Python. Dendritic spine morphology analyses were performed by a technician blind to exposure condition of the rat. Two separate cohorts were conducted for the dendritic spine morphology assessment to allow for test-retest reliability (statistically there were no observed effect of cohort on any of the test parameters).

### 4.9. Effects of Administration of Trichostatin a on AIE-Induced Enhancement of EtOH Microinjected into the Posterior VTA on Dopamine and Glutamate Levels in the AcbSh

The rats were pretreated with 2 mg/kg (ip,) of histone deacetylase (HDAC) inhibitor Trichostatin A (TSA) or vehicle (DMSO in PBS, 1:5 dilution) two days prior to micro-micro and 3 h prior to microinjection of EtOH or aCSF for micro-micro experiment. (TSA was dissolved in DMSO in PBS, 1:5 dilution). This protocol of pretreatment with the toxin TSA has been shown to alter multiple presumed EtOH-related behaviors [54,55]. 

A portion of VTA DA neurons co-release DA and glutamate in the AcbSh [78]. Recently published data has indicated that only co-administration of EtOH + nicotine (but not equivalent EtOH or nicotine alone administrations) directly into the posterior VTA results in the concurrent release of DA and glutamate in the AcbSh [38]. The enhanced DA release in the AcbSh following EtOH microinjection into the posterior VTA in AIE treated rats could be associated with a comparable increase in the release of glutamate in the AcbSh. Therefore, the current experiment examined the effects of TSA to mediate the release of both DA and glutamate in the AcbSh in rats exposed to AIE treatment.

Overall, there were 50 (26 AIE and 24 CON) Wistar males used in this experiment. The experiment design was a 2 (AIE vs. CON) × 2 (EtOH concentration—75 mg% or 150 mg% EtOH) × 2 (HDAC inhibitor—vehicle or TSA) between-subject factorial analysis. The group size for AIE rats was 6–7 rats, while the group size in the CON rats was 5–7 rats.

Samples were collected as indicated above. All samples were immediately frozen on dry ice and stored at −80 °C until analysis. All samples were split and assessed for dopamine and glutamate content with separate high performance liquid chromatography (HPLC) testing.

### 4.10. Effects of Administration of Trichostatin a on AIE-Induced Enhancement of EtOH Reward in the Posterior VTA

Given the results of Experiment 4, the effects of TSA on AIE-induced enhancement of EtOH reward in posterior VTA was determined. AIE treatment reduced the amount of EtOH required to establish self-administration into the posterior VTA in both Wistar and Taconic Alcohol-Preferring (tP) rats [12]. Briefly, in male and female Wistar rats, AIE treated rats self-administered 75 mg% EtOH directly into the posterior VTA, while CON rats did not. Similarly, AIE treated tP male rats self-administered 50 mg% directly into the posterior VTA, while CON tP rats did not [12]. The current experiment examined the effects of TSA to alter the leftward shift in the EtOH dose response curve (EtOH reward) in AIE treated rats.

The rats were pretreated with 22 mM/0.5 μL (same as intra -accumbal dose used in Sprow et al., 2014) TSA two days prior to the initiation of ICSA testing. The same dose of TSA was administered into the posterior VTA prior (3 h) to each ICSA test sessions (acquisition phase—sessions 1–4), extinction—sessions 5–6 aCSF substitution), and reinstatement (session 7—the EtOH is available again). TSA was dissolved in DMSO in PBS, 1:5 dilution). Doses of TSA employed in the current experiment are identical of past reports indicating that TSA administration can counter the effects of AIE [54,55]. 

In male and female Wistar rats, the experiment was a 2 (sex) × 2 (AIE vs. CON) × 2 (EtOH concentration−75 mg% or 150 mg% EtOH) × 2 (HDAC inhibitor—vehicle or TSA) between-subject factorial analysis (*n* = 4–5/group). In tP rats, a single concentration of EtOH was examined (50 mg%). The experimental design was a 2 (sex) × 2 (AIE vs. CON) × 2 (HDAC inhibitor—vehicle or TSA) between-subject factorial analysis (*n* = 4–5/group).

### 4.11. Histologies

Upon termination of Microinjection-Microdialysis and ICSA experiments, a solution of 1% bromophenol blue dye was injected into the infusion site and the animals sacrificed. Brains were removed and immediately frozen at −80 °C, for slicing into 40-µm sections with a cryostat microtome. Slides were stained with cresyl violet and examined for infusion site verification using the atlas of Paxinos and Watson. 

### 4.12. Statistical Analysis

Statistical analyses for the Microinjection-Microdialysis and ICSA data sets were performed through mixed factor ANOVAs (SPSS 2.0). The algorithm of statistical analysis followed the flowchart defined by Keppel and Zedeck [79]. In Experiment 1, the overall analysis was a mixed factor ANOVA with a within-subject factor of ‘Sample Time’ and between-subject factors of ‘Adolescent Exposure’ and ‘EtOH Concentration’. The analysis for Experiment 2 consisted of individual ANOVAs performed for the genetic expression for each individual gene. The data for the dendritic spine morphology experiment (Experiment 3) were the average value for multiple images collected from the same subject (2–5/subject). The analyses for Experiment 3 consisted of individual 2 (sex) × 2 (AIE vs. CON) ANOVAs performed for each spine type across multiple measures. In Experiment 4, the overall analysis was a mixed factor ANOVA with a within-subject factor of ‘Sample Time’ and between-subject factors of ‘Adolescent Exposure’, ‘EtOH Concentration’, and ‘HDAC Exposure’. In Experiment 5, the overall analysis was a mixed factor ANOVA with a within-subject factor of ‘Session’ and between-subject factors of ‘Adolescent Exposure’, ‘EtOH Concentration’, ‘Sex’, and ‘HDAC Exposure’ in each rat strain (no comparison between tP and Wistar rats).

Post-hoc Tukey’s “b” tests were performed to determine group differences. The Tukey’s b post-hoc comparison is a modified Tukey post-hoc comparison that reduces the prohibitive penalty for unequal sample size. There is a single assumption of ANOVAs that cannot be violated. The statistical integrity of ANOVAs is greatly reduced when the assumption of ‘Independence of Measure’ is violated [79]. Therefore, within-subject variables should not be treated like between-subject variables and analyzed with forced ANOVAs and inappropriate post-hoc analyses. Proper within-subject post-hoc analyses are t-tests and orthogonal contrasts [79]. Given the general lack of understanding of orthogonal contrasts, we are reporting only the findings from t-test analyses. Concerns of Type I error rate inflation is eliminated by the proper reading of Rodgers (1967). We adhere to the replicated finding that Type 1 error rate inflations is Ψ*_i_* (*i* = 1, …, *J* − 1). Given the effect size of the current data set Type 1 error rate inflation is not a concern for any reported analyses [80,81]. 

## Figures and Tables

**Figure 1 ijms-22-11733-f001:**
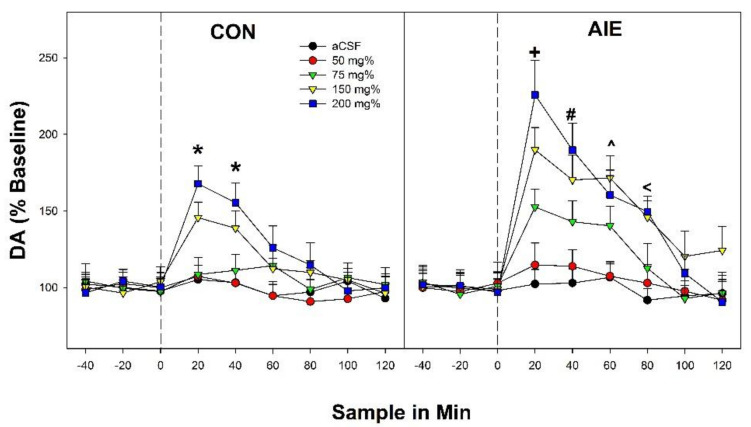
Dopamine levels (%Baseline) in the AcbSh following microinjections of EtOH into the posterior VTA in male Wistar rats that received water treatment (CON; left panel) or binge-EtOH exposure during adolescence (AIE; right panel). * indicates that in CON rats, the 150 and 200 mg% EtOH groups had significantly higher levels of DA in the AcbSh. + indicates that in AIE rats 200 mg% > 150 mg% > 75 mg% > 50 mg% and aCSF, and 75 mg%, 150 mg%, 200 mg% AIE > CON rats. # indicates that in AIE rats 200 mg% and 150 mg% > 75 mg% > 50 mg% and aCSF, and 75 mg%, 150 mg%, 200 mg% AIE > CON rats. ^ indicates that in AIE rats 75 mg%, 150 mg%, 200 mg% > 50 mg% and aCSF, and 75 mg%, 150 mg%, 200 mg% AIE > CON rats. **<** indicates that in AIE rats 150 mg%, 200 mg% > 50 mg%, 75 mg%, and aCSF, and 150 mg%, 200 mg% AIE > CON rats. Data are mean ± SEM.

**Figure 2 ijms-22-11733-f002:**
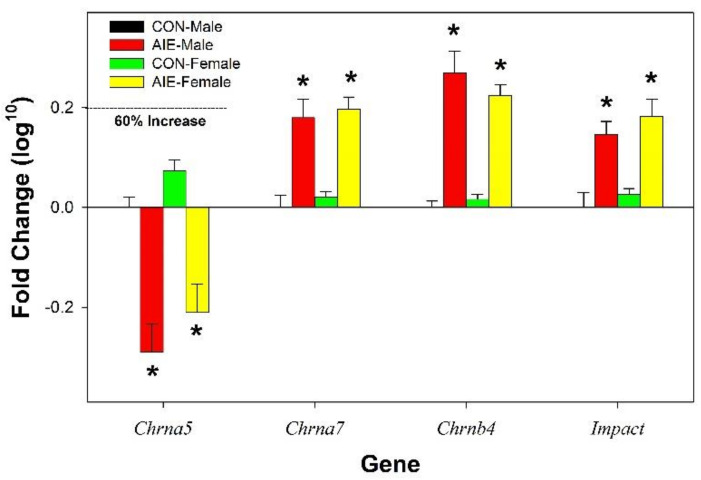
Fold change (log10 normalized to CON males) expression for DA and Ach related genes in the AcbSh in adult CON and AIE male and female Wistar rats. * indicates significantly different from CON-Males and CON-Female values. Data are mean ± SEM.

**Figure 3 ijms-22-11733-f003:**
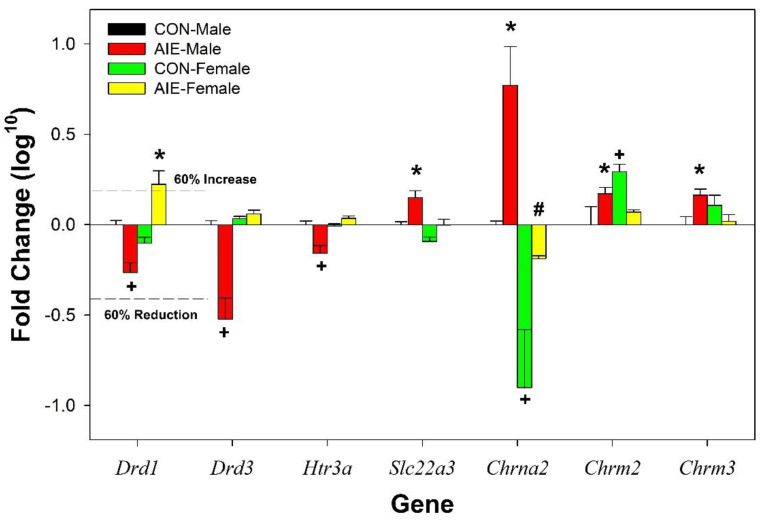
Fold change (log10 normalized to CON males) expression for DA and Ach related genes in the AcbSh in adult CON and AIE male and female Wistar rats. * indicates significantly greater expression compared to CON-Males. + indicates significantly lower expression compared to CON-Males. Data are mean ± SEM.

**Figure 4 ijms-22-11733-f004:**
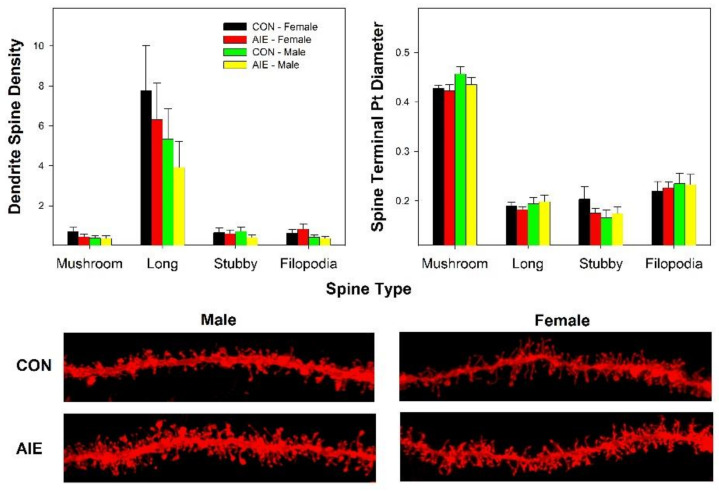
Dendritic Spine Density (left panel) and Spine Terminal Point (Pt) Diameter in the AcbSh in CON and AIE exposed rats during adulthood. Representative images of distal dendritic segments of medium spiny neurons in the AcbSh. There were no significant findings. Data are mean ± SEM.

**Figure 5 ijms-22-11733-f005:**
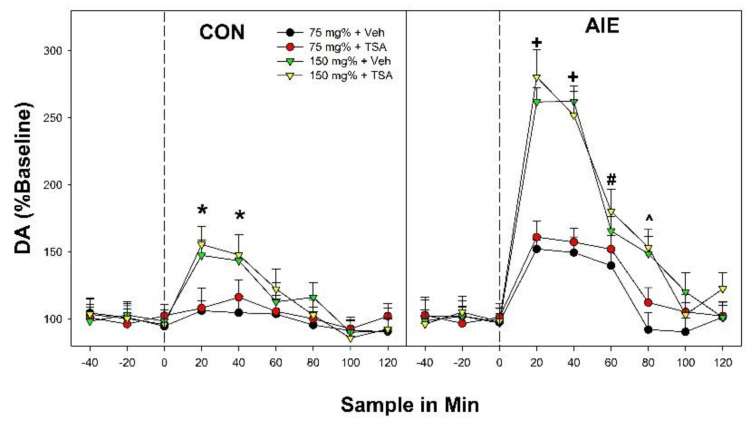
Dopamine levels (%Baseline) in the AcbSh following microinjections of EtOH into the posterior VTA in male Wistar rats that received water treatment (CON; left panel) or binge-EtOH exposure during adolescence (AIE; right panel) following pretreatment/treatment with the HDAC inhibitor Trichostatin A (TSA). TSA failed to alter any measure. * indicates that in CON rats, the 150 mg%-Veh and 150 mg%-TSA groups had significantly higher levels of DA in the AcbSh than 75 mg%-Veh and 75 mg% TSA groups. + indicates that in AIE rats 150 mg%-Veh and 150 mg%-TSA > 75 mg%-Veh and 75 mg%-TSA (all AIE groups different from CON groups). # indicates that all AIE rats (both Veh and TSA) are significantly different from CON groups. ^ indicates that in AIE rats 150 mg%-Veh and 150 mg%-TSA groups are different from all other groups. Data are mean ± SEM.

**Figure 6 ijms-22-11733-f006:**
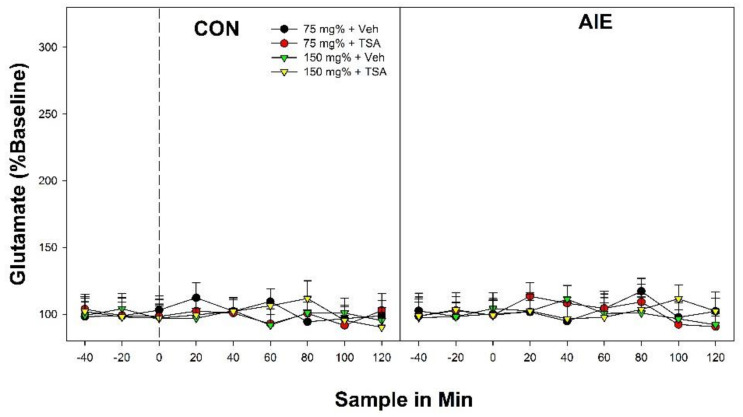
Glutamate levels (%Baseline) in the AcbSh following microinjections of EtOH into the posterior VTA in male Wistar rats that received water treatment (CON; left panel) or binge-EtOH exposure during adolescence (AIE; right panel) following pretreatment/treatment with the HDAC inhibitor Trichostatin A (TSA). There were no observed alterations in glutamate levels in any group. Data are mean ± SEM.

**Figure 7 ijms-22-11733-f007:**
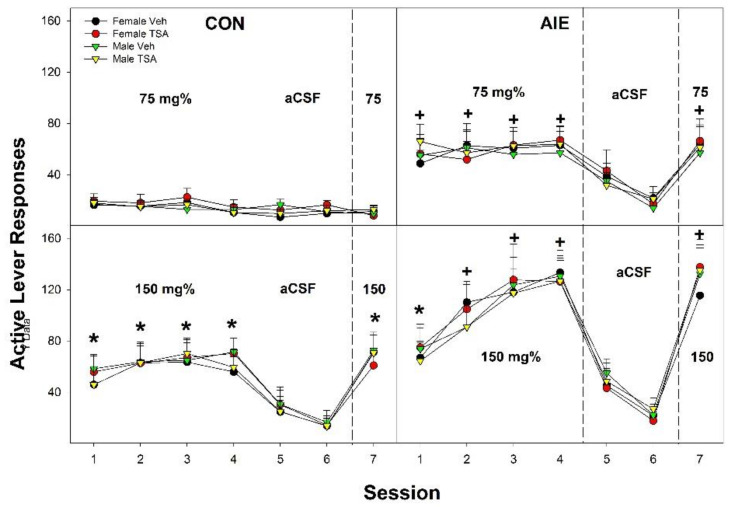
Active lever responses in male and female Wistar exposed to CON or AIE during adolescence and pretreatment/treatment with the HDAC inhibitor Trichostatin A (TSA) before testing for EtOH self-administration into the posterior VTA during adulthood. * indicates lever discrimination between active and inactive lever and significantly more responding than 75 mg% CON rats. + indicates lever discrimination between active and inactive lever and significantly more responding than AIE rats compared to CON rats (at that EtOH concentration. Data are mean ± SEM.

**Figure 8 ijms-22-11733-f008:**
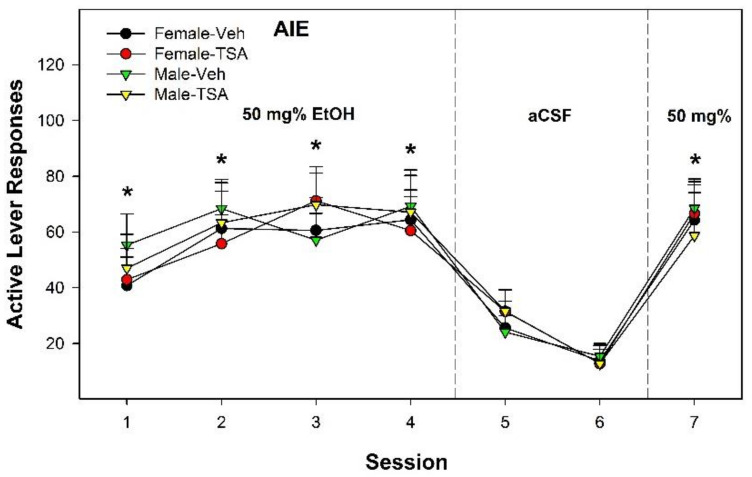
Active lever responses in male and female tP exposed to CON or AIE during adolescence and pretreatment/treatment with the HDAC inhibitor Trichostatin A (TSA) before testing for EtOH self-administration into the posterior VTA during adulthood. * indicates lever discrimination between active and inactive lever. There was no evidence that TSA modified EtOH self-administration into the posterior VTA in tP rats. Data are mean ± SEM.

**Table 1 ijms-22-11733-t001:** Depicts the mean (±SEM) change in gene expression in male and female Wistar AIE and CON rats (log^10^ transformation) for all detected genes in the AcbSh. **Bold** indicates statistical significance (see Figure 2 and Figure 3 for details).

Gene	CON-Male	SEM	AIE-Male	SEM	CON-Fem	SEM	AIE-Fem	SEM
*Ache*	−5.20 × 10^−18^	0.01	−0.06581	0.02	0.02172742	0.02	0.008743	0.02
*Chrm1*	−6.01 × 10^−17^	0.02	−0.27312	0.02	−0.23705888	0.01	−0.0676	0.03
*Chrm2*	−2.24 × 10^−16^	0.1	**0.171**	**0.036**	**0.2933**	**0.041**	0.0701	0.012
*Chrm3*	−3.93 × 10^−17^	0.045	**0.1637**	**0.034**	0.1069	0.056	0.019	0.037
*Chrm4*	−1.69 × 10^−16^	0.04	−0.06339	0.03	−0.06055621	0.03	−0.056317	0.02
*Chrm5*	−3.72 × 10^−17^	0.0213	−0.29	0.056	0.0736	0.021	−0.2103	0.0568
*Chrna2*	−1.67 × 10^−16^	0.021	**0.7713**	**0.214**	**−0.9008**	**0.32**	**−0.1875**	**0.016**
*Chrna3*	−1.98 × 10^−16^	0.05	−0.01599	0.02	0.03186998	0.02	−0.102369	0.04
*Chrna4*	−9.02 × 10^−17^	0.04	−0.08848	0.01	0.03209411	0.03	−0.028314	0.01
*Chrna5*	−3.72 × 10^−17^	0.03	**−0.28998**	**0.02**	0.07362836	0.02	**−0.110304**	**0.02**
*Chrna6*	−3.24 × 10^−17^	0.03	0.000396	0.03	0.04061805	0.03	−0.093764	0.03
*Chrna7*	1.71 × 10^−16^	0.0234	**0.1797**	**0.0362**	0.0199	0.012	**0.1959**	**0.025**
*Chrnb1*	0	0.01	−0.05674	0.03	0.0560767	0.01	−0.036859	0.02
*Chrnb2*	−6.19 × 10^−17^	0.02	−0.0506	0.02	0.02185517	0.02	0.0028137	0.03
*Chrnb3*	0	0.02	−0.06387	0.01	0.04405689	0.03	−0.066709	0.02
*Chrnb4*	−6.01 × 10^−17^	0.0125	**0.2696**	**0.043**	0.016	0.0102	**0.2243**	**0.0214**
*Comt*	−9.77 × 10^−17^	0.03	−0.08802	0.03	−0.06186377	0.02	−0.064171	0.05
*Ddc*	0	0.02	−0.01335	0.02	0.04915982	0.03	−0.07346	0.02
*Drd1*	−1.07 × 10^−16^	0.03	**−0.2645**	**0.053**	−0.1017	0.032	0.2235	0.074
*Drd2*	0	0.02	−0.08844	0.03	0.00745837	0.04	−0.095716	0.03
*Drd3*	−1.01 × 10^−16^	0.0216	**−0.5226**	**0.116**	0.0341	0.012	0.0596	0.021
*Drd5*	−1.01 × 10^−16^	0.03	−0.02727	0.01	0.00479843	0.02	0.0753891	0.04
*Htr3a*	5.09 × 10^−17^	0.021	**−0.1575**	**0.042**	−7.61 × 10^−03^	0.0134	0.0354	0.0125
*Impact*	4.02 × 10^−17^	0.03	**0.1457**	**0.0265**	0.0264	0.0113	**0.1827**	**0.0341**
*Maob*	3.47 × 10^−17^	0.01	0.063724	0.02	−5.38 × 10^−05^	0.03	0.0202155	0.02
*Oprm1*	−1.25 × 10^−16^	0.02	0.043333	0.01	0.06459679	0.01	0.0560767	0.03
*Slc22a3*	9.25 × 10^−17^	0.0164	**0.15**	**0.037**	−0.0926	0.024	−1.88E−03	0.032
*Slc5a7*	−2.59 × 10^−16^	0.01	−0.00635	0.03	−0.1726617	0.04	0.0745436	0.03
*Slc6a3*	1.50 × 10^−17^	0.02	0.053151	0.01	0.02481664	0.03	−0.126964	0.02
*Tbp*	−9.25 × 10^−18^	0.02	0.02016	0.02	−0.05185733	0.02	−0.043138	0.04
*Th*	2.31 × 10^−17^	0.01	−0.01825	0.01	0.00307981	0.02	−0.116577	0.01

## Data Availability

Indiana University has an extensive Data Sharing Plan. All digital data is stored on a general data server. All researchers are required to maintain detail research notes to correspond with the collected data. The goal of the IU Data Sharing Plan is to encourage collaborations within the University and to foster replication in researchers outside the university. The authors hold an extremely high standard of transparency for the collected data. Upon request from outside investigators, the PI will provide the raw data and research notes. In addition, given the general poor understanding of statistics in science, the PI will readily guide researchers on how to properly analyze the data.

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
