# Peer review of "Adolescent Intermittent Ethanol (AIE) Enhances the Dopaminergic Response to Ethanol within the Mesolimbic Pathway during Adulthood: Alterations in Cholinergic/Dopaminergic Genes Expression in the Nucleus Accumbens Shell"

_ijms, 2021, doi:10.3390/ijms222111733_

Round 1

Reviewer 1 Report

Previous studies show that alcohol exposure during adolescence leads to hyper active dopamine system, which increases risk for addiction and other disorders, and the authors seek to provide mechanistic information underlying this alcohol caused hyper dopamine state. The authors show that alcohol injection into posterior VTA causes a greater increase in dopamine but not glutamate release into the NAcb shell in rats exposed to adolescent alcohol. In addition, X genes, which have the capacity to regulate dopamine release, are also increased by adolescent alcohol. However, there were no changes in NAcb spine morphology, with lack of changes in glutamate genes, and lack of effect of HDAC inhibitor (which can regulate synapse formation), together supports hypothesis that adolescent alcohol effects are specific to dopamine. This is an interesting and important series of studies. Comments generally address need for clearing writing, especially to give fuller explanations of many points so that the purpose and interpretation of different experiments is easier to understand, and so that the paper is useful for readers not in this specific field.

P1 The abstract is incomplete and should describe the actual findings more completely. It should mention names of genes impacted by AIE and how it might relates to the greater dopamine increase. Should also mention that AIE did not change glutamate release, which agrees with lack of spine change that adolescent alcohol effects are specific to dopamine. Finally to mention why HDAC inhibitor was tested, and what does lack of effect indicate. These changes would give valuable context in the abstract, rather than a list of seeming disconnected findings.  

Line50 unclear why “however” is needed here, just a second paragraph of alcohol problems with human AIE.

Line 59. Many groups use adolescent alcohol exposure with vapor, or voluntary intake that reaches reasonable BACs, in mice. Thus, to say the “vast majority” of studies have too low exposure is overstated (and no literature is cited to support this).

Line 54 (and line 105). This paragraph thoroughly describe evidence for hyper dopamine, but needs to have some summary about the functional and behavioral importance of hyper dopamine (for readers not in the field). Line 105 mentions this but again needs to say why and how hyper dopamine would differently promote adult behavior.

Line 98 thought to “be” mediated

Line 121 What would TCA be proposed to do (what is the purpose of the study)

Line 196 Since there were sex differences in AIE gene changes, were there sex differences in DA in dialysis

Line 205 Was data collected for glutamate genes but not analyzed? It would be very interesting if there was no AIE effect on glutamate genes, which would validate lack of glutamate dialysis changes or spine morphology changes, emphasizing the specificity of AIE for dopamine (and ACh).

Line 349-382 Aren’t there studies showing ACh receptor regulation of NAcb DA release? Would be important to mention.

Line 384 This paragraph seems unnecessary. Instead, to discuss what behavioral and cellular effects TSA has been shown to inhibit, thus the rationale for testing TSA, and emphasizing that any such mechanisms are not playing a role in AIE caused hyper dopamine. Perhaps then a short discussion of TSA issues that make it a poor human therapy

Lines 419-461 These paragraphs belongs more in a review. The authors present no evidence linking ACh receptors to the hyper dopamine state which seems the main point of the work. Thus, while the points in the paragraph are important, the link is unclear. Might be strengthened here if ACh receptors are known to modulate DA release, even if this is not tested here. Alternately, emphasize ACh changes in the abstract as a major finding of the paper.

Overall, the discussion would be greatly improved by noting what specific future studies would be helpful to shed continuing new information on the important mechanisms of the hyper dopamine state and all other points addressed.

Author Response

First, we would like to thank the reviewer for their comments. Editorial comments were incorporated into the revision, but not indicated in the response to the reviewer.

The journal encourages only 200-word abstracts.  We have attempted to incorporate the suggestions of the reviewer into the limited abstract space.  It is not possible to mention all significant alterations in gene expression within the abstract.

The importance of a hyper-dopaminergic system is now included.

The theory of how TSA works vary within the field.  Primarily, TSA is conceived to have effects by eliminating/reversing the epigenetic factors that were caused by a mediating factor.  In this case, the theory is that AIE produced epigenetic alterations that are the biological basis of altered systems that are observed during adulthood and that administration of TSA reverses these epigenetic factors and returns the organism to the 'natural' state.  That is the theory, but there is little evidence to sufficiently indicate that this effect of TSA (or other epigenetic factors) produces these reversals.

There was no difference in basal levels of DA between the sexes.  This is now indicated in the manuscript.

We did not perform analyze glutamate genes in the current experiment.  We were consistent in examining the same genes in the pVTA (Hauser et al., 2019), AcbSh (current manuscript), and CeA (soon to be published manuscript).  The reviewer is correct, an examination of glutamate genes would be of high interest.

We have added a section indicating that DA release in the AcbSh is regulated by NAchRs.  We thank the reviewer for suggesting this edit, it was needed.

TSA has so many global effects that it would be hard to select specific ones that may mediate the observed effects.

We added some Bo Soderpalm manuscripts and others indicating that NAchRs regulate DA release in the AcbSh.

Future studies section has been added.

Reviewer 2 Report

Summary:

This well thought-out set of studies evaluated the impact an established model of AIE has on persistent changes in DA function in the NAc shell. By probing the impact of AIE on basal DA levels as well as in response to an array of different doses, spanning both binge and super-binge relevant alcohol levels, the authors compellingly demonstrate that AIE is capable of producing long-lasting changes in DA release to ethanol that mirror altered self-administration directly into the NAc shell. These effects are highly relevant to understanding the long-term impact of AIE on later life propensity for developing an AUD and contribute to a growing body of research characterizing the consequences of AIE using the standardized NADIA model (something the field greatly benefits from). In addition, the paper evaluates the efficacy of a currently relevant HDAC inhibitor and calls into question its viability as a potential therapeutic target in the future.

Broad Comments:

            The abstract could benefit from incorporation of the specific methodological details relevant to the AIE.

The introduction draws specific focus to the more recent concepts of high intensity / extreme intensity drinking. A more thorough characterization of what the author’s model is comparable to and differences that have been observed that may be specific to one form of drinking vs another would be useful.

Given the increasing importance that sex differences play in the alcohol literature, particularly in regard to AIE, a brief paragraph touching on observed sex differences in the introduction would inform the choice to incorporate both sexes and potentially the few differences observed later.

The methods discuss the Wistars being derived from an on-site colony however, in the self-administration study Taconic P-rats are included that aren’t discussed in the subjects portion. Given the relevance of early life shipping stress discussed, including specific details regarding the early life history of these animals should be highlighted.

In the Taqman study (experiment 2) the n in a couple of groups is 5 or 6. It would be worthwhile to incorporate power analyses for all studies.

A portion of the discussion was spent exploring polysubstance abuse and the relevance of the ChRNA7 effect observed from experiment 2 however, one of the most interesting effects I noted was the sex effect in ChRNA2. Further exposition on this would be really interesting!

While I thought the discussion was well written, a brief paragraph discussing limitations and alternative interpretation would improve the document. While unavoidable in a study of this nature, single-housing stress, surgery, and isoflurane exposure all have been documented as interacting with AIE/acute ethanol. Similarly, specific discussion of why the TSA dose was selected and whether perhaps administration earlier post-AIE would produce different results would be informative.

Specific Comments:

Lines 475-479. Incorporating relevant information about peak BECs achieved at the beginning of this AIE and by the final ethanol exposure would be valuable.

Author Response

Response to Reviewer 2

We thank the reviewer for their comments.  We have incorporated the statements of the reviewer and believe that the impact of the manuscript has been increased.

Added a brief AIE methodology component into the limited sized abstract.

The AIE model is more comparable to the high intensity drinking observed in humans.  This is now indicated in the manuscript.

We have added a sex differences paragraph to the manuscript.  We observe many sex differences in the CeA, and this will be highlighted in that coming manuscript.

The tP rats were bred on campus and maintained in the colony of the PI.  Thus, Wistar and tP rats were treated identical.  This is now indicated in the manuscript. 

Power analyses were performed and are no indicated in the manuscript.

The Chrna2 sex difference in basal expression is interesting.  We have added a small section highlight this observed effect and others.

We have listed some limitations with the reported data sets.

BECs were not collected because in adolescent rats, the stress of blood extraction has the potential to interact with other variables.